# Estimating the Cost of Production of Two Pentatomids and One Braconid for the Biocontrol of *Spodoptera frugiperda* (Lepidoptera: Noctuidae) in Maize Fields in Florida

**DOI:** 10.3390/insects14020169

**Published:** 2023-02-09

**Authors:** Jermaine D. Perier, Muhammad Haseeb, Daniel Solís, Lambert H. B. Kanga, Jesusa C. Legaspi

**Affiliations:** 1Center for Biological Control, College of Agriculture and Food Sciences, Florida A&M University, Tallahassee, FL 32307, USA; 2Department of Entomology, University of Georgia, Tifton, GA 31794, USA; 3Agribusiness Program, College of Agriculture and Food Sciences, Florida A&M University, Tallahassee, FL 32307, USA; 4Insect Behavior and Biocontrol Research Unit, Center for Medical, Agricultural and Veterinary Entomology, Agricultural Research Service, United States Department of Agriculture, Tallahassee, FL 32308, USA

**Keywords:** maize, fall armyworm, IPM, pentatomid, parasitoid, production, natural enemies, population management, cost analysis

## Abstract

**Simple Summary:**

Natural enemies have long been a tool for pest regulation in agricultural systems and other pest-impacted ecosystems. Despite extensive evaluations of integrated pest management programs, they have provided many different benefits that are yet to be documented. Small-scale growers and farmers stand to benefit the most from effective integrated pest management programs, especially with the increasing failures of cheaper insecticide-based control options. Here, we provide a cost analysis for a small-scale farm, the production which will help growers to promote the use of natural enemies and regional integrated pest management.

**Abstract:**

The fall armyworm is a polyphagous lepidopteran pest that primarily feeds on valuable global crops like maize. Insecticides and transgenic crops have long been a primary option for fall armyworm control, despite growing concerns about transgenic crop resistance inheritance and the rate of insecticide resistance development. Global dissemination of the pest species has highlighted the need for more sustainable approaches to managing overwhelming populations both in their native range and newly introduced regions. As such, integrated pest management programs require more information on natural enemies of the species to make informed planning choices. In this study, we present a cost analysis of the production of three biocontrol agents of the fall armyworm over a year. This model is malleable and aimed towards small-scale growers who might benefit more from an augmentative release of natural enemies than a repetitive use of insecticides, especially since, though the benefits of using either are similar, the biological control option has a lower development cost and is more environmentally sustainable.

## 1. Introduction

The fall armyworm, *Spodoptera frugiperda* (J.E. Smith) (Lepidoptera: Noctuidae), is a native insect to the tropical and sub-tropical regions of the American continent. Historically, the species has established itself as a pest of many regional agricultural and horticultural systems, leading to the excessive use of conventional and transgenic methods for control [1,2]. Sole reliance on these methods eventually created resistant strains of *S. frugiperda*, resulting in many reports of control failures to insecticidal chemistries and transgenic crops [3,4]. In some cases, selection for resistant strains of *S. frugiperda* even led to inheritance of the resistant trait by their offspring [5]. Given the current growing distribution of *S. frugiperda* in the African, Asian and Oceanian regions [6,7,8,9,10,11,12,13,14,15,16,17] and the potential threat to the European continent [18], alternative methods for sustainable control of this pest are greatly needed, as these areas account for more than 80% of the global population [19]. Food security in these geographical areas is at high risk due to the extensive agricultural host range of *S. frugiperda* [20] and the limited documented control options for these regions. Specifically for maize, annual reductions in yield can be higher than 30% [21], with its respective economic losses [8].

Integrated pest management is perhaps the most desirable approach to managing *S. frugiperda*, with identifying and applying natural enemies being a necessary component of any developed program [22]. Despite control differences, organic farming shares similar benefits for using natural enemies, and is often more likely to be adopted [23]. Biological control is a longstanding approach that predates insecticides and regulates the pest population through many agents, such as natural enemies [24,25]. As an ecosystem service, biological control provides many unrecognized benefits, and a constant population presence of natural enemies that can be manipulated to improve its efficacy for pest control [26]. Natural enemies emerged in many cases as an alternative to the adverse effects of conventional control in our ecosystems due to its increased sustainability [25]. A practical method for *S. frugiperda* control in the Americas is biological control agents [27,28,29]. These agents have regulated the species for eons and have coevolved with their pest counterparts. Regardless of human input, natural enemies have substantially controlled *S. frugiperda* in many cropping systems, with at least 50% success rates of *S. frugiperda* mortality [30,31].

*Podisus maculiventris* (Say) and *Euthyrhynchus floridanus* (L.) (Hemiptera: Pentatomidae) are predaceous generalist stinkbugs that feed on lepidopteran pests as well as other insect pests. Another biological control agent, *Cotesia marginiventris* (Cresson) (Hymenoptera: Braconidae), is a common parasitoid emerging from parasitized *S. frugiperda* larvae from field collections [28,32]. All three insects were previously evaluated and shown to be capable of simultaneous use, under the correct timing and life stage conditions, for controlling *S. frugiperda* [29].

In comparison, cost-benefit analyses have shown that biological control methods are cheaper than those conducted on insecticides, regardless of the biocontrol method implemented (classical, augmentative), and with higher chances of success [24]. In addition, it is possible that differences in control may exist when migratory behaviors are taken into consideration for both pest and biocontrol agents, since the immigration and emigration of these species may influence the control outcomes. As to which direction this influence will lean still requires further evaluation at a species level. In addition, biocontrol agents may relocate during foraging. Although foraging events can be tracked with equipment to determine distribution patterns [33], it is critical to include some conservative approaches for natural enemy retention, as these improve available numbers for adequate pest control [34,35,36], especially with biocontrol agents that are slow-acting in the earlier phases of establishment [37]. The overall impact of natural enemies is still poorly known [30]. Thus, their study is warranted, not only to improve our understanding of their efficacy [38], but also to develop cost-effective methods for their production. In this study, we evaluated the cost of establishing a production for three natural enemies of *S. frugiperda* that are native to the sub-tropical southeastern region of the United States. The analysis presented here can easily be modified for other natural enemies, as these predators and parasitoid species may provide the best option for effective *S. frugiperda* population management, especially for small-scale growers.

## 2. Materials and Methods

The production of all predators and parasitoids, as well as the fall armyworm *S. frugiperda,* occurred at the Center for Biological Control at Florida Agricultural, and Mechanical University in Tallahassee, Florida. All colonies were initially collected from existing colonies that had been in rotation for at least two years prior. Colonies of the pentatomids *P. maculiventris* and *E. floridanus*, the parasitoid *C. marginiventris,* and the fall armyworm *S. frugiperda* for this evaluation were reared according to colony establishment protocols described in Perier et al. [29].

This methodology was adapted to determine the cost of production of homegrown natural enemies for small-scale growers. To ensure accuracy, the colonies were reared, and all production costs were documented. The production costs were determined based on the market value of items in the United States in 2019, when the evaluation occurred. Similarly, implementation costs were estimated using recommended distribution records provided by commercial suppliers of the same or similar species at that time.

### 2.1. P. maculiventris and E. floridanus Production

The lifecycles of *P. maculiventris* and *E. floridanus* were initially recorded during colony establishment to determine the length of time and materials used to rear one egg to the adult stage at 26 ± 2 °C, 55 ± 5% RH, and 14:10 (L:D) h photoperiod. For individual evaluations, Petri dishes (150 mm) were lined with filter paper, and a cotton ball (presoaked in water in a smaller Petri dish) was inserted. Five Petri dishes were prepared for the eggs (n = 4) of each predator species. A 30.48 cm^3^ insect cage lined with paper towels, cotton balls (presoaked in water), and pieces of egg cartons inserted for oviposition were prepared for the adults. A total of 20 eggs were reared to adults of each predator species.

Eggs and first instar nymphs of both species were only provided water-soaked cotton balls for humidity and hydration, respectively. However, starting with the second instar, medium-size larvae of the yellow mealworm (*Tenebrio molitor*) were provided for nutrition. Mealworms were provided every two days, with the number of mealworms given at each stage recorded. Water-soaked cotton balls were replaced as needed. This study provided no other feeding materials to the pentatomids, but reports indicate that supplementing prey with pea pods and other substances for nutrition may also positively impact rearing [39]. Following the final molt from the fifth instar, adults were separated and placed in larger containers.

The annual expenditure for producing both pentatomid species was calculated using the following formulas:$$Annual\;Investment\;Cost=\frac{Total\;investment\;cost}{Total\;number\;of\;years\;of\;operation}\;Annual\;Variable\;Cost=Mothly\;variable\;cost\times 12\;months$$

The cost of producing a single pentatomid using this production model was determined by first calculating the total number of insects produced in one year after mortality, and then by calculating the cost of one pentatomid, using the following formulas:$$Total\;insects\;per\;year=Insects\;per\;month\times 12\;\;\;Cost\;of\;one\;insect=\frac{Annual\;variable\;cost\;}{Total\;insects\;per\;year}$$

Calculations were repeated for both *P. maculiventris* and *E. floridanus*, and were used to calculate their respective annual expenditure.

### 2.2. C. marginiventris Production

Production of *C. marginiventris* was dependent on the rearing of its lepidopteran host, *S. frugiperda*; therefore, it was necessary to first establish colonies of *S. frugiperda* prior to production evaluation of the parasitoid. Similarly, production costs for *S. frugiperda* were calculated before that of the parasitoid, and were eventually added to the cost of producing *C. marginiventris*.

Evaluations began with adult *C. marginiventris* kept in a rearing cage and fed 10% honey/sucrose solution. Second instar fall armyworm larvae were used to rear the larval instars of the parasitoid. Because the parasitoid prefers maize-plant-feeding larvae, these second instars were fed maize leaf clippings and maize kernels before the introduction of the parasitoid. Parasitism had to be visually confirmed after the introduction and required exact experimental conditions [29]. Parasitism could take up to 30 min. following the introduction of the host, *S. frugiperda*. Due to the lifecycle of *C. marginiventris*, eggs cannot be separated from the host. Instead, the number of emerged larvae was used to determine the number of eggs laid in the host and assumed to be a 1:1 ratio (one egg = one pupa). Then, the parasitized larvae were stored in 1 oz. cups and fed the Multiple Species Diet until emergence [29].

The annual expenditure to produce *C. marginiventris* was calculated using the following formulas:$$Annual\;Investment\;Cost=\frac{Total\;investment\;cost}{Total\;number\;of\;years\;of\;operation}$$
Annual Variable Cost=[(Monthly variable cost parasitoid+Monthly variable cost FAW)×11 months]+Monthly variable cost FAW 

Similarly, the cost of producing a single parasitoid using this production model was determined by first calculating the total number of *C. marginiventris* produced in one year after mortality and then by calculating the cost of one parasitoid using the following formulas:$$Total\;insects\;per\;year=Insects\;per\;month\times 12\;\;\;Cost\;of\;one\;insect=\frac{Annual\;variable\;cost\;}{Total\;insects\;per\;year}$$

### 2.3. Cost of Implenmenting P. maculiventris, E. floridanus and C. marginiventris for Biocontrol

As previously indicated, we recommended the simultaneous use of these three biological control agents [29] to control the fall armyworm. As such, we analyzed the cost of implementation for each insect, then the cost for all three together.

The individual cost of each insect was determined by multiplying the recommended commercial distribution by the cost of one insect (calculated in Section 2.2 above), and then by the desired distribution area, using the following formula:$$C_{x}=\left(I_{n}\times C_{i}\right)\times D_{i},$$
where

*C_x_* = cost of implementation;*I_n_* = recommended number of insects per area;*D_i_* = production area;*C_i_* = calculated cost of one insect.

For all three together, the cost was determined as follows:$$C_{s}=C_{x}{}_{1}+C_{x}{}_{2}+C_{x}{}_{3}\cdots +C_{x}{}_{n+1,}$$
where
*C_s_* = cost of simultaneous implementation;$C_{x}$ = cost of implementation.

## 3. Results

### 3.1. P. maculiventris and E. floridanus Production

The annual investment cost for producing the pentatomids as biocontrol agents were similar, as both *P. maculiventris* and *E. floridanus* required the same equipment for rearing (Table 1). Our production analyses were done for a production that would last a minimum of 10 years. Therefore, with a total initial investment of USD 486.84 for all equipment, the annual investment cost would be USD 48.68 for a joint production (using the same equipment alternatively for both species) or USD 97.37 for individual equipment for each species.

Our model produced 20 adults of both species at an annual variable cost of USD 183.60 for *P. maculiventris* and USD 367.20 for *E. floridanus* (Table 2). Average mortality during production was only 1%, which meant a 99% survivorship to the adult stage using the rearing protocol of Perier et al. [29]. *Podisus maculiventris* completed its lifecycle from egg to adult within 31 days (one month), while *E. floridanus* required at least twice as long and averaged 60 days for similar full development. As a result, 12 generations of *P. maculiventris* and six generations of *E. floridanus* were produced in our annual production. At 20 adults per generation for each species, the total number of insects produced for the year was 240 *P. maculiventris* adults and 120 *E. floridanus* adults using the production model. As such, a single adult was produced for USD 0.77 and USD 3.06 for *P. maculiventris* and *E. floridanus*, respectively.

### 3.2. C. marginiventris Production

*Cotesia marginiventris* production also depends on its host, which meant a delay in its production to establish *S. frugiperda*. The reported cost of production for *C. marginiventris* also includes the cost of *S. frugiperda* production (Table 3). *S. frugiperda* took approximately 31 days to complete an entire lifecycle. Similarly, from confirmed oviposition, *C. marginiventris* took approximately 31 days to complete its life stages. As such, 12 generations of *S. frugiperda* were produced, and 11 generations of *C. marginiventris* were produced, using this model for the year. One less generation of *C. marginiventris* was produced due to the need to produce its host first, but production began once the 2nd–3rd instar *S. frugiperda* larvae were available for oviposition. A total annual variable cost of USD 284.51 was recorded for *C. marginiventris* production, including a USD 271.2 annual variable cost for *S. frugiperda*. A total of 550 *C. marginiventris* adults were produced for the year from 550 *S. frugiperda* larvae (ratio of 1:1) using this model, with the production of a single adult costing USD 0.52 (Table 4).

### 3.3. Augmentative Biocontrol Implementation

For the augmentative release of these biocontrol agents, we referred to the square footage recommended by Planet Natural (2019), a longstanding organic farming company that mass produces biocontrol agents of similar types. They recommend two predacious stinkbugs per square foot (sq. ft.) and 1 to 3.6 parasitoids per sq. ft. to control caterpillars. The only alteration to this recommendation for our model was the use of 3 parasitoids per sq. ft. As a result, the cost of implementation was USD 1.54 per sq. ft. for *P. maculiventris*, USD 6.12 per sq. ft. for *E. floridanus* and USD 1.56 per sq. ft. for *C. marginiventris*.

## 4. Discussion

Biological control programs often require large numbers of natural enemies when implemented [2] and are often guided by the severity of the pest infestation [40]. Although validation and quality assurance of natural enemies can be lengthy [41], it is often cheaper than an insecticide. Therefore, a cost-effective approach to natural enemy production for species like *S. frugiperda* needs to be budget-friendly and scalable to further reduce costs and promote their use. In addition, the added labor for production potentially increases the region’s economic output by increasing the number of jobs available. Regardless, excluding transgenic crops, pesticides remain the primary choice for pest control, especially as the cost of insecticides gradually decreases with time [42]. For the feasibility of this model, we recommend that supplies be purchased in moderate or bulk proportions to produce biocontrol agents at more competitive prices and lower monthly variable costs.

The economic burden of *S. frugiperda* can be grouped into two categories. First, direct economic impact includes the initial investment cost for establishing a biocontrol program, the rearing cost for natural enemies, the cost of equipment, and even the cost of labor. Direct impacts are influenced by the ongoing market but can be mitigated through the larger purchases of equipment and material, and longer production times. The other category, indirect economic impact, is heavily influenced by the loss of crop yield, which drives the market of the produced goods. Therefore, we target small producers and growers with this model plan, as large-scale growers would likely see more significant benefits from rapid insecticide options paired with multiple layered insect pest management programs.

On the contrary, small-scale farmers might find insecticide options more expensive, especially with increasing control failures amongst cheaper, overused options. As such, we focused on using materials and equipment that could be supplemented with household items or innovative creations that produce the same results. With some preparation, it should be possible to reduce the given investment cost of production without reducing the benefits of using biocontrol agents.

Augmentative biocontrol is in a critical stage of development, despite existing for decades. Modern augmentative biocontrol is already comparable to insecticides, with the benefit of reduced development costs and more sustainable and environmentally friendly approaches [43]. However, multiple releases may be required to maintain the desired control level [24]. As such, many mass-rearing productions require validation of quality assurance [41]. This model produced upwards of 100 adult insects per species for a year. Our small-scale approach was the reason for the number produced, as we focused more on cost evaluation than mass production. However, this model is scalable and could be expanded to reach the desired number of natural enemies. These three biological control agents, *P. maculiventris*, *E. floridanus* and *C. marginiventris,* were simultaneously produced to be used in a joint management program, as previously evaluated [29]. However, growers should consider their agricultural systems when considering these options, as their productions may only support one of these species due to conflicting control methods. It might even be more beneficial production-wise to alternate these species and their releases until the desired establishment.

Our cost analysis identified *C. marginiventris* and *P. maculiventris* as the cheaper production options of this model, USD 1.56 and USD 1.54 per sq. ft., respectively. However, the higher cost of production for *E. floridanus* (USD 6.12 per sq. ft) can be offset by its longevity, which might translate into more extended control periods. Except for *C. marginiventris,* the immature stages of these natural enemies (2nd instar onwards) can feed on the larval stages of the fall armyworm (the most destructive phase). However, augmentative control does require multiple applications. For this model, the exact number of applications required for establishment still needs to be evaluated for these species. Since the model was conducted in an area with already established field populations, perhaps this model might provide an option for the local rearing of native predators and parasitoids, while in continents such as Africa, where 90% of the natural enemies used in insect pest management programs are imported [43], these species can be maintained with this model for multiple applications after the initial purchase.

Natural enemy production for local use might be the best option for small growers and rural farmers [30]. However, the rate of adaption in these communities is still uncertain, and might be plagued by educational and socioeconomic barriers. Also, the cascading effect of the number of offspring generations produced by the initial release of these predators will need to be thoroughly evaluated to deduce the cost-benefit ratio of the three species and the number of applications required for establishment in non-native regions. These prices depend on the current market value of the materials and equipment at the initial point of production, and would therefore require an economic analysis to determine the overall impact on the farming community. However, a communal approach to producing these natural enemies might offset costs even further, and reduce the overall financial burden of development.

## 5. Conclusions

The cost of producing biological control agents varies based on the operation’s size, the materials’ market values, and the demand. *P. maculiventris*, *E. floridanus* and *C. marginiventris* were easily reared using the production method reported at variable costs per sq. ft. with substantial numbers for populating an area. This production method favors small-scale production and community-funded operations. A community production (e.g., farm co-op) would further reduce the calculated costs reported, thus maximizing the intended benefit, and substituting equipment with innovative creations would further reduce these costs.

## Figures and Tables

**Table 1 insects-14-00169-t001:** The initial investment cost for producing 20 *P. maculiventris* and 20 *E. floridanus* eggs to adults.

Equipment	Cost/Unit ^a^	Unit	Total
Insect cages (30.48 cm^3^)	USD 150.94	2	USD 301.88
Petri dishes (150 mm)	USD 0.91	100	USD 90.44
Petri dishes (35 × 10 mm)	USD 0.14	500	USD 67.02
Forceps	USD 11.86	2	USD 23.72
Camel-hair brushes	USD 1.89	2	USD 3.78
Total			USD 486.84

^a^ Cost/Unit subject to the market value of items.

**Table 2 insects-14-00169-t002:** Variable monthly expenditure for producing 20 *P. maculiventris* and 20 *E. floridanus* eggs to adults.

Materials	Cost/Unit ^a^	Unit	Total
Mealworms	USD 0.01	150	USD 2.00
Cotton balls	USD 0.02	200	USD 3.60
Filter paper	USD 0.21	15	USD 3.20
Paper towels	USD 2.00	2	USD 4.00
Water cups	USD 0.07	15	USD 1.00
Egg carton	USD 1.00	1.5	USD 1.50
Total			USD 15.30 ^b^

^a^ Cost/Unit subject to the market value of items. ^b^ Monthly cost should be multiplied by two for *E. floridanus* production.

**Table 3 insects-14-00169-t003:** The initial investment cost for producing *C. marginiventris* and *S. frugiperda* to adults.

Equipment and Materials	Cost/Unit ^a^	Unit	Total
Insect cages (20.5 cm^3^)	USD 123.97	1	USD 123.97
Aspirator	USD 8.99	1	USD 8.99
Aspirator syringe bulb	USD 7.67	1	USD 7.69
Forceps	USD 11.86	1	USD 11.86
Blender	USD 127.52	1	USD 127.52 ^b^
Camel-hair brushes	USD 1.89	2	USD 3.78
Round shape cake pans, 22.86 cm dia. (metal)	USD 2.31	4	USD 9.24
Wire mesh (0.32 cm grid)	USD 27.99	1	USD 27.99
Vermiculite	USD 9.99	2	USD 19.98
Scissors	USD 6.99	1	USD 6.99
Total			USD 348.01

^a^ Cost/Unit subject to the market value of items. ^b^ Insect diet blender.

**Table 4 insects-14-00169-t004:** Variable monthly expenditure for the production of 50 *C. marginiventris* eggs to adults.

Species	Material	Cost/Unit ^a^	Unit	Total
*C. marginiventris*	Kim wipes	USD 0.10	5	USD 0.62
	Cotton balls	USD 0.02	12	USD 0.21
	Honey	USD 0.26	1	USD 0.26
	2oz cups	USD 0.06	2	USD 0.12
Sub-Total				USD 1.21
*S. frugiperda*				
	Multiple species diet	USD 5.41	3	USD 16.23
	Paper towels	USD 2.00	1	USD 2.00
	Ziplock bags	USD 0.09	12	USD 1.07
	1oz cups	USD 0.04	50	USD 1.74
	2oz cups	USD 0.06	4	USD 0.24
	Linseed oil	USD 0.06	0.02	USD 0.32
	Cotton balls	USD 0.02	24	USD 0.48
	Honey	USD 0.26	2	USD 0.52
Sub-Total				USD 22.6
Grand Total				USD 23.81

^a^ Cost/Unit subject to the market value of items.

## Data Availability

The data presented in this study are available on request from the corresponding authors.

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
