# Peer review of "Estimating the Cost of Production of Two Pentatomids and One Braconid for the Biocontrol of Spodoptera frugiperda (Lepidoptera: Noctuidae) in Maize Fields in Florida"

_insects, 2023, doi:10.3390/insects14020169_

Round 1
Reviewer 1 Report
I have read the manuscript thoroughly.
1. Insecticides and transgenic crops have long been a primary option for fall armyworm control despite growing concerns about transgenic crop resistance inheritance and the rate of insecticide resistance development. Global dissemination of the pest species has highlighted the need for more sustainable approaches to managing overwhelming populations both in their native range and newly introduced regions. As such, integrated pest management programs require more information on natural enemies of the species to make informed planning choices. In this study, authors present a cost analysis of the production of three biocontrol agents of the fall armyworm over a year. The model is malleable and aimed towards small-scale growers who might benefit more from an augmentative release of natural enemies than a repetitive use of insecticides.
2. This paper presented the cost analysis of the use natural enemy which is beneficial for small-scale growers.
3. This manuscript is continuation from author's pervious research since there are two related published article (No 21 and 22) in this topic. Author also referred to other related published article by others.
4. For the next work, the cost may be compared to other countries since the cost might be different in some countries that facing similar problem with Fall Army Worm.
5. The conclusion is consistent with the evidence and arguments presented.
6. Most of cited article related to research about Fall Army Worm.
7. Figures are presented well.
In general, this manuscript provides both new and important information. This manuscript has also been well written. I have just found a minor correction in the reference section where some species name must be written in italics such as in reference number 1 - 10, 12, 19, 23, 26 and 19.
Author Response
I have read the manuscript thoroughly.
- Insecticides and transgenic crops have long been a primary option for fall armyworm control despite growing concerns about transgenic crop resistance inheritance and the rate of insecticide resistance development. Global dissemination of the pest species has highlighted the need for more sustainable approaches to managing overwhelming populations both in their native range and newly introduced regions. As such, integrated pest management programs require more information on natural enemies of the species to make informed planning choices. In this study, authors present a cost analysis of the production of three biocontrol agents of the fall armyworm over a year. The model is malleable and aimed towards small-scale growers who might benefit more from an augmentative release of natural enemies than a repetitive use of insecticides.
- This paper presented the cost analysis of the use natural enemy which is beneficial for small-scale growers.
- This manuscript is continuation from author's pervious research since there are two related published article (No 21 and 22) in this topic. Author also referred to other related published article by others.
- For the next work, the cost may be compared to other countries since the cost might be different in some countries that facing similar problem with Fall Army Worm.
- The conclusion is consistent with the evidence and arguments presented.
- Most of cited article related to research about Fall Army Worm.
- Figures are presented well.
In general, this manuscript provides both new and important information. This manuscript has also been well written. I have just found a minor correction in the reference section where some species name must be written in italics such as in reference number 1 - 10, 12, 19, 23, 26 and 19.
Response: As advised, the scientific names of the species have been Italicized.

Reviewer 2 Report
The fall armyworm, Spodoptera frugiperda is a serious, invasive agricultural pest, which is the subject of a global warning from the United Nations Food and Agriculture Organization (FAO). The surprisingly rapid spread of fall armyworm and its significant capacity to generate high-yield losses have attracted increased attention worldwide. Biological control by applying predators and parasitoids is known as an important and effective measure to control fall armyworm. The manuscript estimated the cost of production of three biocontrol agents for the biocontrol of spodoptera frugiperda (lepidoptera: noctuidae) in maize fields in Florida. In general, the manuscript is well organized and meaningful. The topic is relevant in the field of “Recent Advances in Fall Armyworm Research”. However, the paper suffers for several limits comings as detailed in the follows.
1. At the beginning of Introduction, some information on the global economic hazards and rapid spread of S. frugiperda should be provided. This aspect of the manuscript needs to be extended.
2. L47 Given the current growing distribution of S. frugiperda in the African and Asian continents [6-9]. This place should add Oceania. The reference list [6-9] indicates that some relevant and important literatures are not quoted.
Such as,
Stokstad, E. New crop pest takes Africa at lightning speed. Science 2017, 356, 473-474.
Goergen, G.; Kumar, P.L.; Sankung, S.B.; Togola, A.; Tamò, M. First report of outbreaks of the fall armyworm Spodoptera frugiperda (J E Smith) (Lepidoptera, Noctuidae), a new alien invasive pest in West and Central Africa. PLoS ONE 2016, 11, e0165632.
Sharanabasappa; Kalleshwaraswamy, C.M.; Asokan, R.; Swamv, H.M.M.; Marutid, M.S.; Pavithra, H.B.; Hegde, K.; Navi, S.; Prabhu, S.T.; Goergen, G. First report of the fall armyworm, Spodoptera frugiperda (J E Smith) (Lepidoptera: Noctuidae), an alien invasive pest on maize in India. Pest Manag. Hortic. Ecosyst. 2018, 24, 23-29.
Qi, G.J.; Ma, J.; Wan, J.; Ren, Y.L.; McKirdy, S.; Hu, G.; Zhang, Z.F. Source regions of the first immigration of fall armyworm, Spodoptera frugiperda (Lepidoptera: Noctuidae) invading Australia. Insects 2021, 12, 1104., etc.
3. L76-78 In comparison, cost-benefit analyses have shown that biological control methods are cheaper than those conducted on insecticides, regardless of the biocontrol method implemented (classical, augmentative), and with higher chances of success [16]. Indeed, compared with chemical control, biological control method has many advantages. However, for the migratory insects, whether the local natural enemy can control a large number of immigrated populations is also a question.
4. L84 easily be be modified. Here should be an error of repetitive writing ‘be’.
5. L85-86 S. frugiperda population management. Especially for small scale growers. The sentence (Especially for small scale growers) is usually used as adverbial, so it is more reasonable to change the period to comma.
6. L92-93 the pentatomids Podisus maculiventris and Euthyrhynchus floridanus, the parasitoid Cotesia marginiventris. Generally, the Latin scientific name in the article is written in full for the first time. The Latin scientific name should be abbreviated here, similar problems also exist in the following manuscript. Please carefully check the Latin scientific name.
7. L97-98 The costs for production were determined based on the market value of items in 2019 when the evaluation occurred. The market value only indicates the year. I think it also need indicate the location because there are great differences in different regions.
8. L101 L132 L189 L191 L201 L205 L207 Latin scientific name problem.
9. L124-125 total number of insects produced in one year. Whether it is necessary to increase the mortality of insects. The same problem exists in the following papers.
10. L167-169 I want to know how is the effect of using natural enemy control FAW alone? Moreover, the control effect of natural enemy is also affected by environmental conditions, host growth and other factors.
11. L198-199 Average mortality during production was only 1% which meant a 99% survivorship to the adult stage using the rearing protocol of Perier et al. [21. Mortality should be considered in the calculation of total produced number of insects.
12. Table 1 to Table 4: Should labor cost be considered?
13. L237-238 The occurrence degree of fall armyworm directly determines the amount of natural enemies and their control effect. Therefore, only refer to the data of Planet Natural (2019) may not be appropriate.
14. L300-301 Biological control is not necessarily the best choice in some regions of Africa, Asia. As the manuscript discussed, educational and socioeconomic barriers will become barriers to the implementation of biological control, especially for small scale growers. Although the concept of biological control has gained wide recognition, how to apply the biological control measures in some underdeveloped regions is also a difficult problem.
Author Response
The fall armyworm, Spodoptera frugiperda is a serious, invasive agricultural pest, which is the subject of a global warning from the United Nations Food and Agriculture Organization (FAO). The surprisingly rapid spread of fall armyworm and its significant capacity to generate high-yield losses have attracted increased attention worldwide. Biological control by applying predators and parasitoids is known as an important and effective measure to control fall armyworm. The manuscript estimated the cost of production of three biocontrol agents for the biocontrol of spodoptera frugiperda (lepidoptera: noctuidae) in maize fields in Florida. In general, the manuscript is well organized and meaningful. The topic is relevant in the field of “Recent Advances in Fall Armyworm Research”. However, the paper suffers for several limits comings as detailed in the follows.
- At the beginning of Introduction, some information on the global economic hazards and rapid spread of S. frugiperdashould be provided. This aspect of the manuscript needs to be extended.
Response: Additional information has been extended on the global economic impacts and invasion of the pest.
- L47 Given the current growing distribution of S. frugiperdain the African and Asian continents [6-9]. This place should add Oceania. The reference list [6-9] indicates that some relevant and important literatures are not quoted.
Such as,
Stokstad, E. New crop pest takes Africa at lightning speed. Science 2017, 356, 473-474.
Goergen, G.; Kumar, P.L.; Sankung, S.B.; Togola, A.; Tamò, M. First report of outbreaks of the fall armyworm Spodoptera frugiperda (J E Smith) (Lepidoptera, Noctuidae), a new alien invasive pest in West and Central Africa. PLoS ONE 2016, 11, e0165632.
Sharanabasappa; Kalleshwaraswamy, C.M.; Asokan, R.; Swamv, H.M.M.; Marutid, M.S.; Pavithra, H.B.; Hegde, K.; Navi, S.; Prabhu, S.T.; Goergen, G. First report of the fall armyworm, Spodoptera frugiperda (J E Smith) (Lepidoptera: Noctuidae), an alien invasive pest on maize in India. Pest Manag. Hortic. Ecosyst. 2018, 24, 23-29.
Qi, G.J.; Ma, J.; Wan, J.; Ren, Y.L.; McKirdy, S.; Hu, G.; Zhang, Z.F. Source regions of the first immigration of fall armyworm, Spodoptera frugiperda (Lepidoptera: Noctuidae) invading Australia. Insects 2021, 12, 1104., etc.
Response: Thank you, the above references have been added including additional references that help to highlight the distribution of the fall armyworm.
- L76-78 In comparison, cost-benefit analyses have shown that biological control methods are cheaper than those conducted on insecticides, regardless of the biocontrol method implemented (classical, augmentative), and with higher chances of success [16]. Indeed, compared with chemical control, biological control method has many advantages. However, for the migratory insects, whether the local natural enemy can control a large number of immigrated populations is also a question.
Response: The risk of migratory behavior of pest on local or augmented populations of natural enemies is an excellent question for evaluation. In this regards, further explanation has been added in the text.
- L84 easily be modified. Here should be an error of repetitive writing ‘be’.
Response: Repetitive word has been removed.
- L85-86 S. frugiperda population management. Especially for small scale growers. The sentence (Especially for small scale growers) is usually used as adverbial, so it is more reasonable to change the period to comma.
Response: The sentence has been modified.
- L92-93 the pentatomids Podisus maculiventris and Euthyrhynchus floridanus, the parasitoid Cotesia marginiventris. Generally, the Latin scientific name in the article is written in full for the first time. The Latin scientific name should be abbreviated here, similar problems also exist in the following manuscript. Please carefully check the Latin scientific name.
Response: The Latin scientific names have been checked and abbreviated.
- L97-98 The costs for production were determined based on the market value of items in 2019 when the evaluation occurred. The market value only indicates the year. I think it also need indicate the location because there are great differences in different regions.
Response: The United States was added as the region as some items were also purchase online from other states, such as the multispecies diet.
- L101 L132 L189 L191 L201 L205 L207 Latin scientific name problem.
Response: Scientific names have been abbreviated.
- L124-125 total number of insects produced in one year. Whether it is necessary to increase the mortality of insects. The same problem exists in the following papers.
Response: The mortality was accounted for in the total calculation. We have now included this statement for further clarity.
- L167-169 I want to know how is the effect of using natural enemy control FAW alone? Moreover, the control effect of natural enemy is also affected by environmental conditions, host growth and other factors.
Response: The previous laboratory study highlights the interactions of the species simultaneously as well as their individual preferences. We agree that it is possible for environmental conditions to impact responses, which is why we recommended further evaluations of this nature. However, for this aspect, the augmentative release of these insects examined would include substantial numbers that would offer adequate control on a cooperative level.
- L198-199 Average mortality during production was only 1% which meant a 99% survivorship to the adult stage using the rearing protocol of Perier et al. [21. Mortality should be considered in the calculation of total produced number of insects.
Yes, the mortality was included in the calculation. A statement has been added to clarify this.
- Table 1 to Table 4: Should labor cost be considered?
Response: The labor cost is of great importance, especially with established production systems. However, it is possible for small growers to reduce expenditure by rearing these insects within their households (family). As such, we only focused on the cost associated with the insect and not employment as a different variable.
- L237-238 The occurrence degree of fall armyworm directly determines the amount of natural enemies and their control effect. Therefore, only refer to the data of Planet Natural (2019) may not be appropriate.
Response: Plant Natural was the only provider of similar agents during the period of the study for this region. As such their measurement were used to estimate our own costs.
- L300-301 Biological control is not necessarily the best choice in some regions of Africa, Asia. As the manuscript discussed, educational and socioeconomic barriers will become barriers to the implementation of biological control, especially for small scale growers. Although the concept of biological control has gained wide recognition, how to apply the biological control measures in some underdeveloped regions is also a difficult problem.
Response: Many barriers exist that might hinder application in some areas. Our aim was to provide the information needed for better adoption and possible community dissemination. Perhaps in areas that prove difficult for rearing, rearing might be administered by agricultural agents that also share valuable crop protection information.

Reviewer 3 Report
In this work, authors developed a study to estimate the production costs of three natural enemies and the relative cost to implement an augmentative biological control program. The work is relevant and may help growers and other people interesting in rear these insects to implement viable colonies of this species.
In general, the work is well structured and conducted. In my opinion, there are some minor points that need to be improved for a more clear presentation of procedures (including rearing of insects and calculations) and to facilitate the understanding of the results obtained and costs/values presented.
I recommend to authors a more detailed description of rearing procedures and material used in each step and a more complete cost study because in my opinion many important items were not included in the production cost estimation (see some comments in the pdf file of the original manuscript).

Author Response
In this work, authors developed a study to estimate the production costs of three natural enemies and the relative cost to implement an augmentative biological control program. The work is relevant and may help growers and other people interesting in rear these insects to implement viable colonies of this species.
In general, the work is well structured and conducted. In my opinion, there are some minor points that need to be improved for a more clear presentation of procedures (including rearing of insects and calculations) and to facilitate the understanding of the results obtained and costs/values presented.
I recommend to authors a more detailed description of rearing procedures and material used in each step and a more complete cost study because in my opinion many important items were not included in the production cost estimation (see some comments in the pdf file of the original manuscript).
Response: More detailed description has been provided.
L45 This phrase is not clear reorganize
Response: The sentence was restructured.
L54-56 italic:
Response: The scientific names were Italicized.
L107 how many eggs in each Petri dish
Response: There were a total of 20 eggs and 4 eggs per Petri dish.
L209 In this cost (investment) was not include the infrastructure (insectary, refrigerators, equipment for rearing room climatization, etc.). Other basic supplies as cleaning materials and others also were not included
Response: Inclusion of these items would benefit larger scale productions that might fall outside the scope of the analysis. By focusing on the basic equipment and materials needed it makes the model more malleable for all levels of production.
L221 not italic here
Response: As advised, the scientific name was Italicized.

Round 2
Reviewer 2 Report
This is my second review of this paper, and I can say it is much improved. It's easier to understand because the authors diligently tried to address the comments of the reviewers and editors. I think the manuscript is meaningful and the topic is relevant in the field of “Recent Advances in Fall Armyworm Research”. So, I recommend this manuscript to publish in < insects >.
Some obvious problems, such as lack of page number and volume number exist in the references. Please check and revise carefully.